# Large Amplitude Vibration of FG-GPL Reinforced Conical Shell Panels on Elastic Foundation

**DOI:** 10.3390/ma16176056

**Published:** 2023-09-03

**Authors:** Jin-Rae Cho

**Affiliations:** Department of Naval Architecture and Ocean Engineering, Hongik University, Jochiwon, Sejong 30016, Republic of Korea; jrcho@hongik.ac.kr; Tel.: +82-44-860-2546

**Keywords:** GPL-reinforced composite, functionally graded, conical shell panel, Pasternak-type foundation, nonlinear free vibration, 2-D meshfree method

## Abstract

Functionally graded (FG) composite structures reinforced by graphene platelets (GPL) have been widely adopted as a state-of-the-art structural element due to their preeminent properties and functional designability. However, most studies are confined to beams, plates, and cylindrical panels, relying on the numerical differential quadrature method (DQM) and the finite element numerical method. In this context, the current study intends to investigate the nonlinear free vibration of FG-GPL-reinforced composite (RC) conical panels resting on an elastic medium by developing a 2-D planar meshfree method-based nonlinear numerical method. The nonlinear free vibration problem is expressed by the first-order shell deformation theory and the von-Kármán nonlinearity. The complex conical neutral surface of the panel is transformed into a 2-D rectangular plane to avoid painstaking mathematical manipulation. The troublesome shear-membrane locking is suppressed by employing the MITC3+shell element, and the derived nonlinear modal equations are solved by introducing a three-step direct iterative scheme. The present method is compared with the DQM through the benchmark experiment, from which a good agreement between the two methods is observed. And, the nonlinear free vibration characteristics of FG-GPLRC conical panels on an elastic foundation are profoundly investigated, and it is found that those are significantly influenced by the foundation stiffness, the amount and dispersion pattern of GPLs, the panel geometry sizes, and the boundary condition.

## 1. Introduction

The notion of functionally graded material (FGM) was introduced in the late 1980s to develop advanced heat-resisting composites for the space shuttle [1]. The traditional lamination-type heat-resisting composites could not successfully protect the space shuttle from the enormous thermal shock while reentering the atmosphere. This crucial defect stemmed from the discontinuity in the material composition distribution across the layer interface [2,3], because the severe thermal stress concentration induced by the material composition discontinuity triggered the micro-cracking and debonding [4,5]. In FGMs, the material discontinuity completely disappeared with the introduction of the graded layer, in which the material composition distribution varies continuously along one or two specific directions [6]. Moreover, the graded material distribution can be purposefully tailored according to the desired function [7], which gave birth to the term functionally graded. Due to its continuity and functionality, the notion of FGM has been extended to various engineering and science fields and is not restricted to the state-of-the-art heat-resisting composite [8,9].

Recently, the notion of FGM has been actively applied to carbon nanocomposites such as carbon graphene platelet- and carbon nanotube-reinforced polymer composites. Graphene platelets (GPLs) and carbon nanotubes (CNTs) have attracted great attention as advanced nanofiller materials due to their extraordinary physical, chemical, and electromagnetic properties [10,11,12]. It has been reported that the properties of carbon nanocomposite are epochally improved when only a small amount of GPLs or CNTs is inserted [13,14]. However, the reinforcement of these advanced nanofillers is limited to the low-weight fraction owing to their high cost, which in turn naturally led to the notion of FGM. In functionally graded GPL- and CNT-reinforced composites, nanofillers are dispersed in a specific pattern through the thickness, and several primitive functional patterns, such as uniform, FG-V, FG-O, FG-X, and FG-Λ were suggested. Since the thermomechanical responses of GPL- and CNT-reinforced composites are substantially affected by the functional pattern, their efforts on the mechanical properties leading to static bending and free vibration have been extensively studied [15,16].

Restricted to the functionally graded GPL-reinforced composites (FG-GPLRC), Gholami and Ansari [17] solved the geometically nonlinear deflection of FG-GPLRC plates subjected to uniform and transverse mechanical loadings by applying the generalized DQM to the sinusoidal SDT. Van Do and Lee [18] explored the bending and free vibration responses of FG-GPLRC cylindrical panels by applying Bezier extraction-based isogeometric analysis (IGA) to the first-order SDT. Gao et al. [19] studied the probalistic stability characteristics of FG-GPLRC beams by considering the multidimensional probability distributions using a non-inclusive Chebyshev metamodel. Nematollahi et al. [20] analytically investigated the nonlinear vibration behavior of thick FG-GPLRC sandwich beams by combining a power-law-based GPL distribution with a higher-order laminated model. Ansari et al. [21] numerically investigated the postbuckling and free vibration of postbuckled porous FG-GPLRC plates via a variational mixed formulation using the third-order shear deformation theory (SDT). Jamalabadi et al. [22] solved the nonlinear vibration of FG-GPLRC conical panels in the elastic medium by applying the 2-D DQM to the first-order SDT. Javani et al. [23] solved the nonlinear free vibration of the FG-GPLRC plate by applying the generalized DQM to the first-order SDT. Mohd and Talha [24] conducted the free vibration analysis of FG-GPLRC porous arches subjected to thermal loading by applying FEM to the higher-order SDT. Garg et al. [25] analyzed the static bending and free vibration of multilayered FG-GPLRC beams using the parabolic function-based SDT and investigated the effects of the amount and functional pattern of GPLs. Cho [26] investigated the free vibration of FG-GPLRC porous cylindrical panels by applying the 2-D natural element method (NEM) to the first-order SDT.

The literature survey informed us that the majority of studies on FG-GPLRC structures were limited to beams, plates, and cylindrical panels by applying DQM or FEM to the SDT. The studies on the conical panels were relatively poor, and the numerical study on the nonlinear free vibration by the meshfree method has been rarely presented. Compared to beam, plate, and cylindrical panels, the geometry configuration of conical panels is identified by more parameters, and its elastic behavior is more complex to investigate. And the non-planar conical surface makes the numerical modeling more painstaking and the numerical accuracy more difficult to secure against membrane locking [27,28] when the standard iso-parametric FEM is used. In this context, the curved neutral surface of the conical panel is transformed to the rectangular plane in this study, and the displacement field is approximated by 2-D NEM, a recently introduced meshfree method [29,30], which is characterized by high-smooth Laplace interpolation functions. The effective material properties of the FG-GPLRC panel are estimated by the Halphin-Tsai formula [31], and the nonlinear free vibration is expressed by the first-order SDT incorporated with the von-Kármán-type geometry nonlinearity. The resulting nonlinear modal equations are solved by the three-step direct iterative algorithm [32], and the inherent shear-membrane locking is held back by the MITC3+shell element [33].

The numerical results are compared with those obtained by the DQM through the benchmark experiment, from which a good agreement is seen between the two methods. Next, the nonlinear free vibration responses of FG-GPLRC conical panels on a Pasternak-type elastic foundation are parametrically and profoundly investigated. It is found that the nonlinearity in free vibration is significantly influenced by the foundation stiffness, the mass fraction and functional pattern of GPLs, the panel geometry dimensions, and the boundary condition. Following the introduction, the nonlinear vibration problem of the FG-GPLRC conical panel, together with the evaluation of effective material properties, are described in Section 2. In Section 3, the NEM approximation using the curve-plane geometry transformation and the locking-free shell element are addressed. The numerical results are presented and discussed in Section 4, and the final conclusion is summarized in Section 5.

## 2. Modeling of FG-GPLRC Conical Shell Panel

Figure 1a depicts a conical shell panel resting on a Pasternak-type elastic medium in which graphene platelets (GPLs) are reinforced, where a coordinate system (x,θ,z) is situated on the panel’s neutral-surface ϖ. The 3-D geometry of a conical panel is characterized by a small radius $R_{1}$, semi-vertex angle α, subtended angle $\theta_{0}$, thickness h, and length L. The radius R(x) of conical panel is expressed by $R\left(x\right)=R_{1}+xsin\alpha$ in the direction of the shell axis. The foundation is completely bonded to the panel without separation, and its load-deflection relationship is expressed by [34]
(1)$$p=k_{w}w-k_{s}\nabla_{\varpi }^{2}w$$
with the Laplace operator $\nabla_{\varpi }^{2}=\partial^{2}/\partial x^{2}+\partial^{2}/R^{2}\partial \theta^{2}$, p the force per unit area, w the panel deflection, $k_{w}$ the Winkler foundation stiffness, and $k_{s}$ the shearing layer stiffness. Graphene platelets in the panel are distributed in a specific functional pattern through the thickness. Four patterns studied in this work are depicted in Figure 1b, where GPLs are uniformly distributed in FG-U, while those are rich at the mid-surface in FG-O, at the bottom surface in FG-Λ, and at the top and bottom surfaces in FG-X, respectively.

Letting $V_{m}\left(z\right)$ and $V_{GPL}\left(z\right)$ be the volume fractions of the underlying matrix and GPLs, then both must obey the physical constraint given by
(2)$$V_{GPL}\left(z\right)+V_{m}\left(z\right)=1$$
where the GPL volume fraction $V_{GPL}\left(z\right)$ is expressed as different thickness functions given by
(3)$$V_{GPL}\left(z\right)=\{\begin{array}{ll} V_{GPL}^{*}, & \mathrm{FG}\;-\;U\;\\ 2\left(1-2\left|z\right|/h\right)V_{GPL}^{*}, & \mathrm{FG}-O\;\\ 2\left(2\left|z\right|/h\right)V_{GPL}^{*}, & \mathrm{FG}-X\\ \left(1-2z/h\right)V_{GPL}^{*}, & \mathrm{FG}-\Lambda \end{array}$$
depending on the GPL distribution pattern. Where $V_{GPL}^{*}$ in Equation (3), is calculated by [35]
(4)$$V_{GPL}^{*}=\frac{g_{GPL}}{g_{GPL}+\rho_{GPL}\left(1-g_{GPL}\right)/\rho_{m}}$$
with the GPL mass fraction $g_{GPL}$ and the densities $\rho_{GPL}$ and $\rho_{m}$ of GPL and the underlying matrix.

GPLs act as an effective rectangular solid fiber, and their geometric configurations are characterized by length $l_{GPL}$, width $w_{GPL}$, and thickness $t_{GPL}$. And the graphene-reinforced composites (GPLRC) with randomly oriented discontinuously short fibers show planar isotropic behavior. For these nanocomposites, the semi-emperical Halphin-Tsai micromechanical model provides better results than the mechanics of materials approach [31]. According to this model, the effective Young’s modulus $E_{C}$ of GPLRC is estimated as [36]
(5)$$E_{C}=\frac{E_{m}}{8}\left\{3\left(\frac{1+\xi_{L}\eta_{L}V_{GPL}}{1-\eta_{L}V_{GPL}}\right)+5\left(\frac{1+\xi_{T}\eta_{T}V_{GPL}}{1-\eta_{T}V_{GPL}}\right)\right\}$$
with
(6)$$\eta_{L}=\frac{E_{GPL}-E_{m}}{E_{GPL}+\xi_{L}E_{m}},\;\eta_{T}=\frac{E_{GPL}-E_{m}}{E_{GPL}+\xi_{T}E_{m}}$$

Here, $E_{m}$ and $E_{GPL}$ are the Young’s moduli of the underlying matrix and GPLs, and $\xi_{L}$ and $\xi_{T}$ denote the geometric parameters defined by
(7)$$\xi_{L}=\frac{2l_{GPL}}{t_{GPL}},\;\xi_{T}=\frac{2w_{GPL}}{t_{GPL}}$$

Meanwhile, the effective density $\rho_{C}$ and the effective Poisson’s ratio $\nu_{C}$ are calculated by
(8)$$\rho_{C}=V_{GPL}\rho_{GPL}+V_{m}\rho_{m}$$
(9)$$\nu_{C}=V_{GPL}\nu_{GPL}+V_{m}\nu_{m}$$
using the linear rule of mixture [7] with $V_{GPL}$ and $V_{m}$ defined in Equation (2).

The current study adopts the first-order shear deformation shell theory, and the displacement $u={\left\{u,v,w\right\}}^{T}$ of conical panel is expressed as
(10)$${\left\{\begin{array}{c} u\\ v\\ w \end{array}\right\}}_{\left(x,\theta ,z\right)}={\left\{\begin{array}{c} u_{0}\\ v_{0}\\ w_{0} \end{array}\right\}}_{\left(x,\theta \right)}+z\cdot {\left\{\begin{array}{c} \beta_{x}\\ \beta_{y}\\ 0 \end{array}\right\}}_{\left(x,\theta \right)}$$
with $d={\left(u_{0},v_{0},w_{0},\beta_{x},\beta_{y}\right)}^{T}$ being the displacement component vector defined on the neutral-surface of the conical panel. By adopting a von Kármán-type geometry nonlinearity to represent the large deflection of the panel, one has the strain-displacement relations given by
(11)$$\begin{array}{lll} \left\{\begin{array}{c} \varepsilon_{xx}\\ \varepsilon_{\theta \theta }\\ 2\varepsilon_{x\theta } \end{array}\right\}=\varepsilon & = & \left\{\begin{array}{c} \frac{\partial u_{0}}{\partial x}+\frac{1}{2}{\tilde{w}}_{,x}\frac{\partial w}{\partial x}\\ \frac{1}{R}\frac{\partial v_{0}}{\partial \theta }+\frac{u_{0}sin\alpha }{R}+\frac{w_{0}cos\alpha }{R}+\frac{1}{2R^{2}}{\tilde{w}}_{,\theta }\frac{\partial w}{\partial \theta }\\ \frac{1}{R}\frac{\partial u_{0}}{\partial \theta }+\frac{\partial v_{0}}{\partial x}-\frac{v_{0}sin\alpha }{R}+\frac{1}{2R}\left({\tilde{w}}_{,x}\frac{\partial w}{\partial \theta }+{\tilde{w}}_{,\theta }\frac{\partial w}{\partial x}\right) \end{array}\right\}+z\cdot \left\{\begin{array}{c} \frac{\partial \beta_{x}}{\partial x}\\ \frac{1}{R}\frac{\partial \beta_{y}}{\partial \theta }+\frac{\beta_{x}sin\alpha }{R}\\ \frac{1}{R}\frac{\partial \beta_{x}}{\partial \theta }+\frac{\partial \beta_{y}}{\partial x}-\frac{\beta_{y}sin\alpha }{R} \end{array}\right\}\\ & = & \left(H_{L}+H_{NL}\right)d \end{array}$$
(12)$$\left\{\begin{array}{c} \gamma_{\theta z}\\ \gamma_{xz} \end{array}\right\}=\gamma =\left\{\begin{array}{c} \beta_{y}+\frac{1}{R}\frac{\partial w_{0}}{\partial \theta }-\frac{v_{0}}{R}cos\alpha \\ \beta_{x}+\frac{\partial w_{0}}{\partial x} \end{array}\right\}=H_{s}d$$

In the co-ordinate system (x,θ,z). Here, $H_{L},\;H_{NL}$, and $H_{s}$ indicate (3×5) and (2×5) gradient-like matrices defined by
(13)$$H_{L}=\left[\begin{array}{ccccc} H_{x} & 0 & 0 & z\cdot H_{x} & 0\\ RS & H_{\theta } & RC & z\cdot RS & z\cdot H_{\theta }\\ H_{\theta } & \left(H_{x}-RS\right) & 0 & z\cdot H_{\theta } & z\cdot \left(H_{x}-RS\right) \end{array}\right]$$
(14)$$H_{NL}=\left[\begin{array}{ccccc} 0 & 0 & {\tilde{w}}_{,x}^{}H_{x}/2 & 0 & 0\\ 0 & 0 & \left({\tilde{w}}_{,\theta }^{}/R\right)\;H_{\theta }/2 & 0 & 0\\ 0 & 0 & \left[{\tilde{w}}_{,x}^{}H_{\theta }+\left({\tilde{w}}_{,\theta }^{}/R\right)\;H_{x}\right]/2 & 0 & 0 \end{array}\right]$$
(15)$$H_{s}=\left[\begin{array}{ccccc} 0 & -RC & H_{\theta } & 0 & 1\\ 0 & 0 & H_{x} & 1 & 0 \end{array}\right]$$
with $H_{x}=\partial /\partial x,\;H_{\theta }=\partial /R\partial \theta ,\;RS=sin\alpha /R$, and RC=cosα/R. Furthermore, the constitutive equations are expressed as [26]
(16)$$\left\{\begin{array}{c} \sigma_{xx}\\ \sigma_{\theta \theta }\\ \sigma_{x\theta } \end{array}\right\}=\sigma =\left[\begin{array}{ccc} Q_{11} & Q_{12} & 0\\ Q_{12} & Q_{11} & 0\\ 0 & 0 & Q_{66} \end{array}\right]\left\{\begin{array}{c} \varepsilon_{xx}\\ \varepsilon_{\theta \theta }\\ 2\varepsilon_{x\theta } \end{array}\right\}=D\left(H_{L}+H_{NL}\right)d$$
(17)$$\left\{\begin{array}{c} \tau_{\theta_{z}}\\ \tau_{xz} \end{array}\right\}=\tau =\left[\begin{array}{cc} Q_{44} & 0\\ 0 & Q_{44} \end{array}\right]\left\{\begin{array}{c} \gamma_{\theta z}\\ \gamma_{xz} \end{array}\right\}=D_{s}H_{s}d$$

With $Q_{11}=E_{C}/\left(1-\nu_{C}^{2}\right),\;Q_{12}=\nu_{C}Q_{11}$, and $Q_{66}=Q_{44}=E_{C}/2\left(1+\nu_{C}\right)$. Two material constant matrices, D and $D_{s}$, are needed to be changed when the problem is extended to nonlinear elasticity [37].

## 3. Analysis of Nonlinear Vibration Using 2-D Meshfree Method

Referring to Figure 2, a NEM grid ${ℑ}_{C}$ is generated on the neutral surface ϖ of conical shell panel by dividing the surface into M Delaunay triangles. Where all vertices of triangles become the nodes of the NEM grid, and the total number of nodes is denoted by N. A curvilinear coordinate (x,s) is used to identify the location on the neutral surface according to the relation of s=Rθ. Next, for the NE approximation, the approximate displacement field $u^{h}\left(x,s,z\right)$ is expressed as
(18)$${\left\{\begin{array}{c} u^{h}\\ v^{h}\\ w^{h} \end{array}\right\}}_{\left(x,s,z\right)}=\sum_{J=1}^{N}{\left\{\begin{array}{c} u_{0}\\ v_{0}\\ w_{0} \end{array}\right\}}_{J}\psi_{J}\left(x,s\right)+\sum_{J=1}^{N}z\cdot {\left\{\begin{array}{c} \beta_{x}\\ \beta_{y}\\ 0 \end{array}\right\}}_{J}\psi_{J}\left(x,s\right)$$
in terms of Laplace interpolation (L/I) functions $\psi_{J}\left(x,s\right)$ [29,30] and the nodal displacement vector $d_{J}={\left(u_{0},v_{0},w_{0},\beta_{x},\beta_{y}\right)}_{J}^{T}$ at node J.

Meanwhile, the mathematical derivation of L/I functions and their manipulation on the conical neutral-surface ϖ is rather complicated. To resolve this problem, the physical NEM grid ${ℑ}_{C}=\left[0,L\right]\times \left[-s_{L},s_{R}\right]$ on the shell neutral surface is transferred to the computational 2-D planar NEM grid ${ℑ}_{R}=\left[0,L\right]\times \left[-\theta_{0}/2,\theta_{0}/2\right]$ using the geometry transformation $T_{C}^{}$ defined by
(19)$$T_{C}^{}:\;\left(x,s\right)\in {ℑ}_{C}\in {ℑ}_{R}\;\to \;\left(\zeta_{1},\zeta_{2}\right)\in {ℑ}_{R}$$
where
(20)$$x=\zeta_{1},\;s=\left(R_{1}+\zeta_{1}sin\alpha \right)\;\zeta_{2}$$

Laplace interpolation functions $\phi_{J}\left(\zeta_{1},\zeta_{2}\right)$ are derived on the 2-D rectangular NEM grid ${ℑ}_{R}$ and those are mapped to the physical NEM grid ${ℑ}_{C}$ through the inverse transformation $T_{C}^{-1}$. From the relations given in Equation (20), the inverse Jacobi matrix $J^{-1}$ is derived as
(21)$$J^{-1}=\left[\begin{array}{cc} \partial \zeta_{1}/\partial x & \partial \zeta_{1}/\partial s\\ \partial \zeta_{2}/\partial x & \partial \zeta_{2}/\partial s \end{array}\right]=\frac{1}{R\left(\zeta_{1}\right)}\left[\begin{array}{cc} R\left(\zeta_{1}\right) & 0\\ -\zeta_{2}\cdot sin\alpha & 1 \end{array}\right]$$

And the partial derivatives $H_{x}$ and $H_{\theta }$ in Equations (13)–(15) are switched to
(22)$$\frac{\partial }{\partial s}=H_{\theta }=\frac{\partial \zeta_{1}}{\partial s}\frac{\partial }{\partial \zeta_{1}}+\frac{\partial \zeta_{2}}{\partial s}\frac{\partial }{\partial \zeta_{2}}=\frac{1}{R\left(\zeta_{1}\right)}\frac{\partial }{\partial \zeta_{2}}=H_{2}$$
(23)$$\frac{\partial }{\partial x}=H_{x}=\frac{\partial \zeta_{1}}{\partial x}\frac{\partial }{\partial \zeta_{1}}+\frac{\partial \zeta_{2}}{\partial x}\frac{\partial }{\partial \zeta_{2}}=\frac{\partial }{\partial \zeta_{1}}-\frac{\zeta_{2}sin\alpha }{R\left(\zeta_{1}\right)}\cdot \frac{\partial }{\partial \zeta_{2}}=H_{1}-\zeta_{2}sin\alpha \cdot H_{2}$$
according to the chain rule.

Introducing Equations (22) and (23) into Equations (13)–(15) results in ${\hat{H}}_{L},\;{\hat{H}}_{NL}$ and ${\hat{H}}_{s}$ with $H_{1}$ and $H_{2}$ instead of $H_{x}$ and $H_{\theta }$:(24)$$T_{C}^{}:H_{L},H_{NL},H_{s}\;\to \;{\hat{H}}_{L},{\hat{H}}_{NL},{\hat{H}}_{s}$$

And the NE approximation of the in-plane strain ε in Equation (11) and the transverse shear (T/S) strain γ in Equation (12) ends up with
(25)$$\varepsilon^{h}=\sum_{J=1}^{N}\left({\hat{H}}_{L}+{\hat{H}}_{NL}\right)\phi_{J}d_{J}=\sum_{J=1}^{N}\left(B_{L}^{J}+B_{NL}^{J}\right)d_{J}$$
(26)$$\gamma^{h}=\sum_{J=1}^{N}{\hat{H}}_{s}\phi_{J}d_{J}=\sum_{J=1}^{N}B_{s}^{J}d_{J}$$

Here, the standard NE approximation (26) of the T/S strain γ using $C^{0}-$L/I functions $\phi_{J}$ may cause shear-membrane locking [27,28] with a big numerical approximation error. One way to avoid this phenomenon is to indirectly approximate the T/S strains using the concept of the MITC3+shell element [33], as addressed in the Appendix. The analytic derivation of Equations (A1) and (A2) in Appendix A using Equations (12) and (26), together with the chain rule between the physical and master coordinates (x,s) and (ξ,η), ends up with
(27)$${\hat{\gamma }}_{}^{e}={\hat{B}}_{e}^{}d_{e}$$

Here, ${\hat{B}}_{e}^{}$ are the (2×15) matrices expressed by ξ,η,z and R, and $d_{e}=\left\{d_{1}^{e},d_{2}^{e},d_{3}^{e}\right\}$ are the (15×1) element-wise nodal vectors.

Meanwhile, the dynamic form of the virtual energy principle for FG-CNTRC conical shell panels on Winkler-Pasternak foundation in Equation (1) is expressed as follows [38]
(28)$$\int_{-h/2}^{h/2}\int_{\varpi }\left[{\left(\delta \varepsilon \right)}^{T}D\varepsilon +{\left(\delta \gamma \right)}^{T}D_{s}\gamma +\delta w\;\left(k_{w}w-k_{s}\nabla_{\varpi }^{2}w\right)\right]d\varpi dz+\int_{-h/2}^{h/2}\int_{\varpi }{\left(\delta d\right)}^{T}m\ddot{d}\;d\varpi \;dz=0$$

In which a (5×5) symmetric matrix m is given by
(29)$$m=\rho \left[\begin{array}{cc} I & m_{1}^{T}\\ m_{1} & m_{2} \end{array}\right],\;m_{1}=\left[\begin{array}{ccc} z & 0 & 0\\ 0 & z & 0 \end{array}\right]$$
with $m_{2}=diag\left(z^{2},z^{2}\right)$ and the (3×3) identity matrix I. Assuming the shell panel is in harmonic motion $d=\bar{d}\cdot e^{j\omega t}$ and introducing Equations (25) and (27) into Equation (28) through the strain-stress relations (17) and (18), the following non-linear modal equation is derived
(30)$$\left[\left(K_{L,\sigma }+\sum_{e=1}^{M}K_{L,s}^{e}+K_{L,ef}\right)+K_{NL}\right]\;\bar{d}\;-\omega^{2}M\bar{d}=0$$

Here, $\omega_{I}$ and ${\bar{d}}_{I}$ are the non-linear natural frequencies and natural modes, and the stiffness matrices (three linear and one non-linear) and the mass matrix are defined by
(31)$$K_{L,\sigma }=\int_{-h/2}^{h/2}\int_{\varpi }B_{L}^{T}DB_{L}\;d\varpi dz$$
(32)$$K_{L,s}^{e}=\int_{-h/2}^{h/2}\int_{\varpi_{e}}{\hat{B}}_{e}^{T}{\hat{D}}_{s}{\hat{B}}_{e}\;d\varpi dz$$
(33)$$K_{L,ef}=\int_{\varpi }\left[k_{w}\Phi_{w}^{T}\Phi_{w}+k_{s}\left(\nabla_{\varpi }\Phi \cdot \nabla_{\varpi }\Phi \right)\right]\;d\varpi$$
(34)$$K_{NL}=\int_{-h/2}^{h/2}\int_{\varpi }\left[B_{L}^{T}DB_{NL}+B_{NL}^{T}DB_{L}+B_{NL}^{T}DB_{NL}\right]\;d\varpi dz$$
(35)$$M=\int_{-h/2}^{h/2}\int_{\varpi }\Phi^{T}m\Phi \;d\varpi dz$$
in terms of $\bar{d}=\left[d_{1},d_{2},\cdots ,d_{N}\right]$, $B=\left[B_{1},B_{2},\cdots ,B_{N}\right]$, $\Phi =\left[\Phi_{1},\Phi_{2},\cdot \cdot \cdot ,\Phi_{N}\right]$ with $\Phi_{I}=diag\left[\phi_{I},\phi_{I},\phi_{I},\phi_{I},\phi_{I}\right]$, $\Phi_{w}=\left[\Phi_{w1},\Phi_{w2},\cdot \cdot \cdot ,\Phi_{wN}\right]$ with $\Phi_{wJ}=diag\left[0,0,\phi_{J},0,0\right]$. In addition, the modified shear modulus matrix ${\hat{D}}_{s}$ is defined by
(36)$${\hat{D}}_{s}=\frac{\kappa }{1+\vartheta \cdot {\left(L_{e}/h\right)}^{2}}\left[\begin{array}{cc} Q_{44} & 0\\ 0 & Q_{44} \end{array}\right]$$
with the shear correction factor κ=5/6, the largest side length $L_{e}$ of triangular element, and a positive shear stabilization parameter ϑ(ϑ>0) [39,40]. The value of ϑ is decided through the preliminary experiment.

The non-linear eigenvalue of equation (30) is solved by the three-step direct iterative scheme [32], which was introduced for plate-like structures. The iteration is terminated when the relative difference between two natural frequencies solved at two consecutive iterations is less than 0.1%.

## 4. Results and Discussion

The numerical experiments are divided into benchmark tests for justifying the present nonlinear numerical method and parametric ones for investigating the nonlinear free vibration characteristics. The matrix of composite conical panels is epoxy, and its material properties are $E_{m}=3\;\mathrm{GPa},$
$\nu_{m}=0.34$, and $\rho_{m}=1.2\;g/{\mathrm{cm}}^{3}$. The average geometry dimensions of GPLs are $l_{GPL}=2.5\;\mu m,$ $w_{GPL}=1.5\;\mu m$, and $t_{GPL}=1.5\;\mathrm{nm}$, and the material properties are $E_{GPL}=1.01\;\mathrm{TPa},$
$\nu_{GPL}=0.186$, and $\rho_{GPL}=1.06\;g/{\mathrm{cm}}^{3}$, respectively. The stiffness and mass matrices in Equations (31)–(35) are computed using 7 Gauss points, except for $K_{L,s}^{e}$, which is integrated using 1 Gauss point. Referring to Figure 3, a 2-D rectangular NEM grid with uniform density 15×15 is taken, and the six lowest natural modes were extracted by Lanczos transformation and Jacobi methods for the whole numerical experiment. The stabilization parameter ϑ was taken by 0.05–0.3 depending on the GPL functional pattern. The boundary conditions specified for the panel edges are simply-supported (S), clamped (c), and free, where S and C are enforced as
(37)$$S:v_{0}=w_{0}=\beta_{y}=0$$
(38)$$C:u_{0}=v_{0}=w_{0}=\beta_{x}=\beta_{y}=0$$

The first benchmark example is a FG-GPLRC conical panel with the geometry dimensions given by $R_{1}/h=8,$
$L/R_{1}=5$ and $\theta_{0}=120^{o}$. Referring to Figure 2, four edges ①, ②, ③, and ④ are subjected to clamped, simply-supported, clamped and simply-supported, which is simply denoted by CSCS in this paper. The fundamental frequency $\omega_{1}$ is calibrated as $\Omega_{L}=\omega_{1}R_{1}/h\cdot \sqrt{\rho_{m}/E_{m}}$, and it was computed by changing the helix angle α, the GPL mass fraction $g_{GPL}^{*}$ and the functional pattern of GPL. Note that the corresponding volume fraction $V_{GPL}^{*}$ can be calculated using Equation (4). The computed non-dimensional frequencies are recorded in Table 1 and compared with those obtained by Jamalabadi et al. [23] using the DQM. When compared with those of the DQM, the present method provides higher frequencies for $\alpha =20^{{}^{\circ}}$ but lower frequencies for $\alpha =40^{{}^{\circ}}$. However, the difference is not significant, such that the maximum relative difference is 3.159%. Note that the foundation stiffness values $k_{w}$ and $k_{s}$ are calibrated as $K_{w}=k_{w}R_{2}^{4}/D_{11}$ and $K_{s}=k_{s}R_{2}^{2}/D_{11}$, where $D_{11}$ is defined by
(39)$$D_{11}=\int_{-h/2}^{h/2}Q_{11}z^{2}dz$$

The geometry dimensions of the second benchmark example without an elastic foundation are $R_{1}/h=10\left(R_{1}=0.1\;m,\;h=0.01\;m\right)$ and $\alpha =45^{{}^{\circ}}$, and the mass fraction $g_{GPL}^{*}$ and functional pattern are 0.8% and FG-X, respectively. The nonlinear-linear frequency (NLF) ratios $\Omega_{NL}/\Omega_{L}$ are computed for two different subtended angles $\theta_{0}$ and for two different boundary conditions. The ratios are computed by changing the non-dimensional peak deflection $W_{\max}^{*}=w_{\max}/h$ from 0.3 to 1.5, and the computed values are compared to those of the DQM [23] in Table 2 and Figure 4. When compared with the DQM, the present method produces higher ratios for $\theta_{0}=90^{{}^{\circ}}$ but lower ratios for $\theta_{0}=180^{{}^{\circ}}$. However, it is clearly seen that the two methods are in good agreement, with the maximum relative difference equal to 3.993%. Thus, it has been verified that the present method accurately computes the nonlinear natural frequencies even with a 2-D coarse planar NEM grid.

Next, the nonlinear free vibration of FG-GPLRC conical panels on an elastic medium is parametrically investigated. The simulation parameters are taken by $R_{1}/h=10,\;\alpha =45^{{}^{\circ}},$ $\theta_{0}=90^{{}^{\circ}}$, $g_{GPL}^{*}=0.8\%$, $K_{w}=300$ and $K_{s}=15,$ unless otherwise specified. Figure 5a shows the effect of the foundation stiffness on the NLF ratio $\Omega_{NL}/\Omega_{L}$, for which the functional pattern and the boundary condition are taken by FG-X and CCCC. It is seen that the ratio $\Omega_{NL}/\Omega_{L}$ uniformly decreases in proportion to the foundation stiffness $\left(K_{w},K_{s}\right)$, because the foundation tends to suppress the radial deflection w of conical panel. In other words, the foundation stiffness which is independent of $W_{\max}^{*}$ reduces the nonlinearity in panel-free vibration. Figure 5b represents the effect of the GPL mass fraction $g_{GPL}^{*}$ on the NLF ratio $\Omega_{NL}/\Omega_{L}$. It is observed that the ratio $\Omega_{NL}/\Omega_{L}$ slightly decreases and becomes saturated with increasing the value of $g_{GPL}^{*}$ nevertheless the panel stiffness increases with increasing the value of $g_{GPL}^{*}$. This reverse trend was caused by the calibration of $k_{w}$ and $k_{s}$ with $D_{11}$, because these two absolute stiffness values become larger as $D_{11}$ increases with $g_{GPL}^{*}$, for the given calibrated stiffness values $K_{w}=300$ and $K_{s}=15.$

Figure 6a represents the effect of the GPL functional pattern on the NLF ratio $\Omega_{NL}/\Omega_{L}$ when the boundary condition is CCCC. The order in the magnitude of $\Omega_{NL}/\Omega_{L}$ is shown to be FG-O > FG-Λ > FG-U > FG-X, and this relative order agrees with one of Jamalabadi et al. [23]. This relative order can be explained by the fact that the magnitude of $D_{11}$ in Equation (39) depends on the vertical distribution of $Q_{11}$ which is influenced by the GPL functional pattern even though the GPL mass fraction $g_{GPL}^{*}$ is kept the same. And, the increase of $D_{11}$ automatically leads to the increase of $k_{w}$ and $k_{s}$ even though $K_{w}$ and $K_{s}$ are fixed as 300 and 15. Here, the magnitude order of $D_{11}$ is FG-X > FG-U > FG-Λ > FG-O, and this relative order is the same for $k_{w}$ and $k_{s}$. Thus, the relative order shown in Figure 6a between the GPL functional patterns can be explained since the ratio $\Omega_{NL}/\Omega_{L}$ decreases as the values of $k_{w}$ and $k_{s}$ become larger, as represented in Figure 5a.

Figure 6b shows the effect of boundary conditions on the NLF ratio $\Omega_{NL}/\Omega_{L}$ when the GPL functional pattern is FG-U. Keep in mind that the combination of four capital letters (e.g., SCSC) denotes a set of boundary conditions specified for four sides ①, ②, ③, and ④ of conical panel shown in Figure 2. One can realize from the comparison that the magnitude of $\Omega_{NL}/\Omega_{L}$ becomes larger when two curved edges ① and ③ of panel are simply-supported and two straight edges ② and ④ are clamped. It is because the nonlinearity in the natural modes, which are bended along the circumferential direction, is inferred to increase when two sides ② and ④ are clamped while the other sides ① and ③ are simply-supported. For the free boundary, FCFC exhibits the lowest level, while it was found that CFCF produces the level between CCCC and CSCS.

Figure 7a represents the effect of relative thickness $R_{1}/h$ on the nonlinear free vibration of the simply-supported FG-Λ conical panel with $R_{1}=0.1\;m$. It is observed that the ratio $\Omega_{NL}/\Omega_{L}$ decreases as the panel thickness becomes smaller, such that the frequency-deflection curve becomes almost linear with the increasing the value of $R_{1}/h$. It is because the panel stiffness becomes smaller in reverse proportion to the panel thickness. Figure 7b represents the effect of subtended angle $\theta_{0}$, where the nonlinear-linear ratio $\Omega_{NL}/\Omega_{L}$ decreases in proportion to $\theta_{0}$ until the value of $\theta_{0}$ reaches to $135^{{}^{\circ}}$. But, thereafter, the ratio increases with increasing the value of $\theta_{0}$, and this trend is confirmed to be consistent with the result of Jamalabadi et al. [23]. The decrease of $\Omega_{NL}/\Omega_{L}$ proportional to $\theta_{0}$ is explained by the fact that the panel stiffness becomes smaller as $\theta_{0}$ increases. It was observed that the variation trend of the frequency-deflection curve with respect to the value of $\theta_{0}$ is slightly influenced by the semi-vertex angle α, as presented in a paper by Jamalabadi et al. [23].

Figure 8a represents the effect of the semi-vertex angle α on the NLF ratio $\Omega_{NL}/\Omega_{L}$ when the GPL functional pattern is FG-O and the boundary condition is CCCC. The axial length ℓcosα is kept unchanged by 0.1 m, and the radius $R_{2}$ used in the calibration of $k_{w}$ and $k_{s}$ is replaced with $R_{1}$ in order to exclude the influence owing to the change of $R_{2}$. It is observed that the ratio $\Omega_{NL}/\Omega_{L}$ uniformly decreases in proportion to the value of α, because the panel stiffness decreases in proportion to α. Figure 8b shows the effect of the axial length of the panel on the nonlinear free vibration of a clamped FG-O conical panel. Note that the panel axial length ℓcosα becomes larger in proportion to the ratio of $R_{2}/R_{1}$ when $R_{1}$ and α are kept unchanged. It is observed that the ratio $\Omega_{NL}/\Omega_{L}$ uniformly decreases proportional to the ratio $R_{2}/R_{1}$, because the panel stiffness becomes smaller with increasing the value of $R_{2}/R_{1}$.

## 5. Conclusions

The nonlinear free vibration of FG-GPLRC conical panels resting on a Pasternak foundation was parametrically investigated using a meshfree-based nonlinear numerical method. For the sake of locking-free, reliable, and effective nonlinear computation, a geometry transformation between the shell surface and the rectangular NEM grid, the MITC3+shell element, and a three-step direct iteration scheme were integrated in the framework of 2-D NEM. Both benchmark and parametric experiments were performed to verify the developed numerical method and to profoundly investigate the nonlinear vibration response. The following key observations were obtained from the numerical results:
The numerical method accurately and effectively computes the nonlinear natural frequencies even using 2-D coarse planar NEM grids;The proposed method shows good agreement with the DQ discretization method, with the maximum relative differences equal to 3.159% in the linear fundamental frequencies and 3.993% in the nonlinear fundamental frequencies;The NLF ratio $\Omega_{NL}/\Omega_{L}$ uniformly decreases in proportion to the foundation stiffness and the GPL mass fraction owing to the deflection suppression and the influence on the foundation stiffness calibration;The GPL function pattern and the boundary condition influence the ratio $\Omega_{NL}/\Omega_{L}$, but their effects are not significant. The magnitude order of $\Omega_{NL}/\Omega_{L}$ due to the former is FG-O > FG-Λ > FG-U > FG-X, and one due to the latter is SCSC > SSSS > CCCC > CSCS;The NLF ratio $\Omega_{NL}/\Omega_{L}$ shows a remarkable uniform decrease proportional to the thickness ratio $R_{1}/h$, the semi-vertex angle α and the axial length $R_{2}/R_{1}$ because the panel stiffness becomes smaller with increasing the values of these three parameters;The subtended angle $\theta_{0}$ has a somewhat peculiar effect on the nonlinear vibration such that the frequency ratio $\Omega_{NL}/\Omega_{L}$ decreases in proportion to $\theta_{0}$ up to the certain value of $\theta_{0}$ but thereafter increases with $\theta_{0}$.

The proposed numerical method accurately and effectively solved the large-amplitude vibration of FG-GPLRC conical panels on an elastic foundation. However, the proposed method considered the geometrical nonlinearity of the linear elastic material model. So, the extension of the present method to nonlinear elasticity would be worthwhile, and this represents a topic that deserves future work.

## Figures and Tables

**Figure 1 materials-16-06056-f001:**
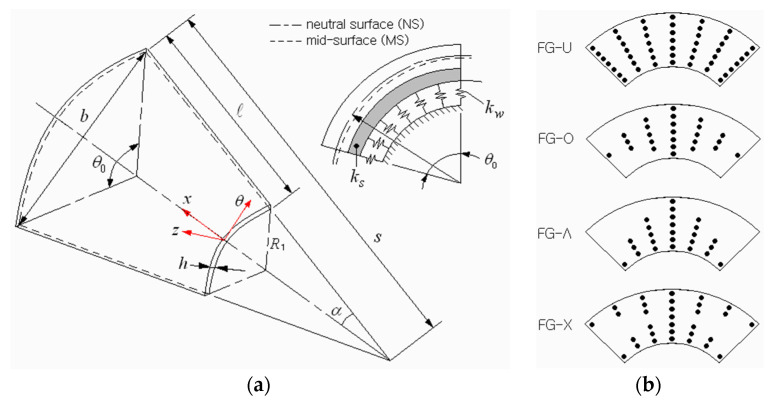
FG-GPLRC conical shell panel: (**a**) the geometry dimensions; (**b**) functional dispersion patterns of GPLs.

**Figure 2 materials-16-06056-f002:**
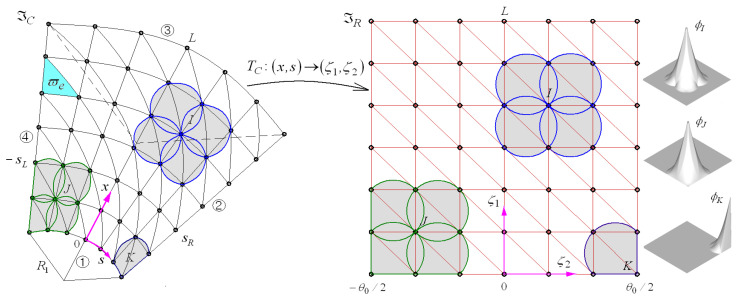
A geometry transformation $T_{C}^{}$ between the physical and computational NEM grids and L/I functions $\phi_{J}\left(\zeta_{1},\zeta_{2}\right)$ defined on 2-D rectangular plane.

**Figure 3 materials-16-06056-f003:**
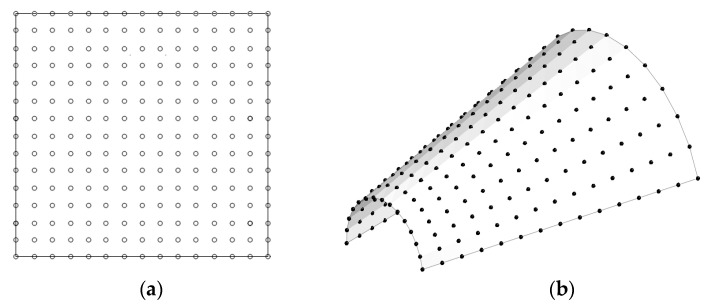
Representation: (**a**) a 2-D uniform 15×15 planar NEM grid; (**b**) its virtual grid mapped to the neutral surface of conical shell panel.

**Figure 4 materials-16-06056-f004:**
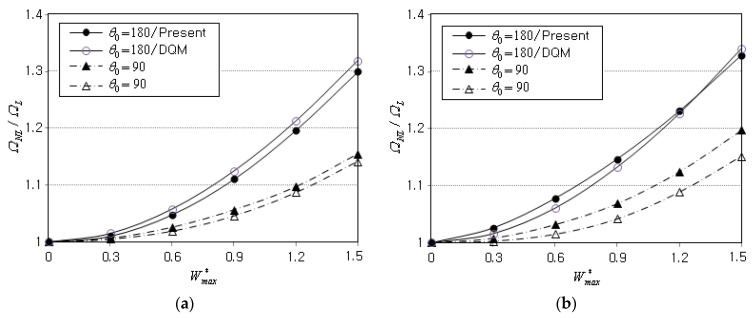
Comparison of frequency-deflection curves of FG-GPLRC conical panels ($R_{1}/h=10,\;\;\;\alpha =45^{{}^{\circ}},$
$g_{GPL}^{*}=0.8\%,\;\mathrm{FG}-X,\;\;\;K_{w}=K_{s}=0$): (**a**) for CCCC; (**b**) for SSSS.

**Figure 5 materials-16-06056-f005:**
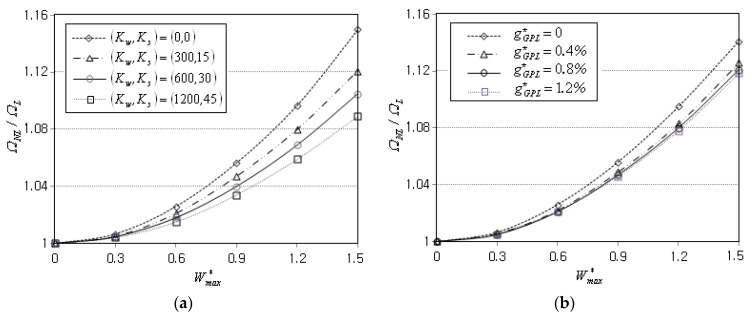
Variation of NLF ratios $\Omega_{NL}/\Omega_{L}$ (FG-X, CCCC): (**a**) to the foundation stiffness; (**b**) to the GPL mass fraction $g_{GPL}^{*}$.

**Figure 6 materials-16-06056-f006:**
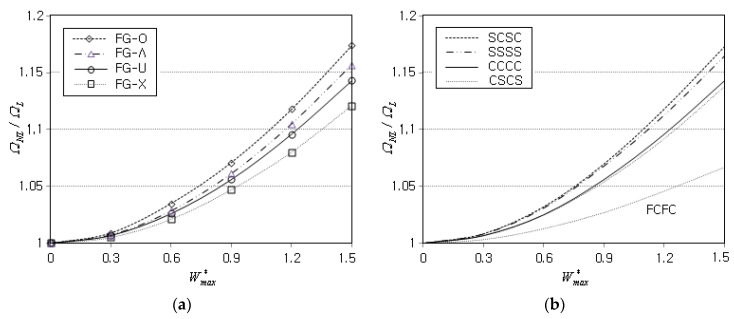
Variation of NLF ratio $\Omega_{NL}/\Omega_{L}$: (**a**) to the GPL functional pattern (CCCC); (**b**) to the boundary condition (FG-U).

**Figure 7 materials-16-06056-f007:**
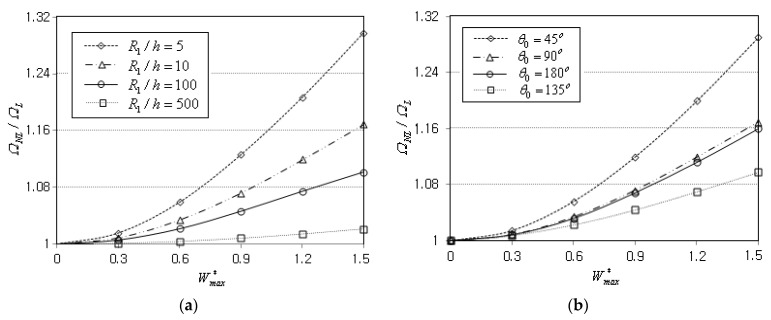
Variation of the NLF ratio $\Omega_{NL}/\Omega_{L}$ (FG-Λ, SSSS): (**a**) to the thickness ratio $R_{1}/h\;\left(R_{1}=0.1\;m\right)$; (**b**) to the subtended angle $\theta_{0}$.

**Figure 8 materials-16-06056-f008:**
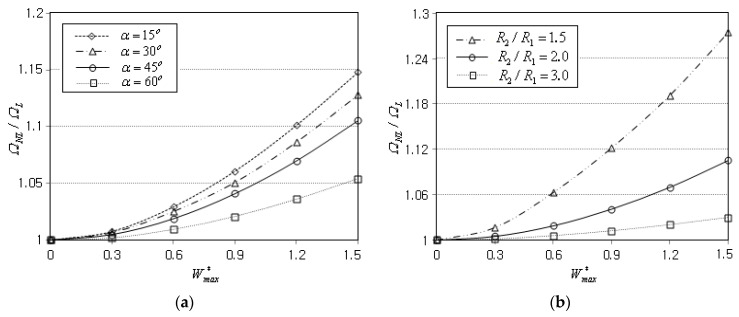
Variation of the NLF ratio $\Omega_{NL}/\Omega_{L}$ (FG-O, CCCC): (**a**) to the semi-vertex angle α (ℓcosα=0.1); (**b**) to the axial length of panel $\left(R_{1}=0.1\;m\right)$.

**Table 1 materials-16-06056-t001:** Comparison of the non-dimensional fundamental frequencies $\Omega_{L}$ of FG-GPLRC conical panel (CSCS, $R_{1}/h=8,$
$L/R_{1}=5,$
$\theta_{0}=120^{{}^{\circ}},\;K_{w}=K_{s}=0$).

Method	α(deg)	$g_{GPL}^{*}$ (%)	GPL Distribution Pattern			
			FG-U	FG-O	FG-A	FG-X
DQM [23]	20	0.4	3.200	2.993	3.131	3.382
		0.8	4.012	3.672	3.873	4.298
	40	0.4	2.299	2.130	2.248	2.412
		0.8	2.881	2.585	2.767	3.059
Present	20	0.4	3.240	3.053	3.187	3.388
		0.8	4.062	3.788	3.968	4.279
	40	0.4	2.236	2.063	2.181	2.372
		0.8	2.803	2.556	2.710	2.999

**Table 2 materials-16-06056-t002:** Comparison of NLF ratios $\Omega_{NL}/\Omega_{L}$ for different subtended angles $\left(R_{1}/h=10,\alpha =45^{{}^{\circ}},g_{GPL}=0.8\%,\;\mathrm{FG}-X,\;K_{w}=K_{s}=0\right)$.

$\theta_{0}\left(deg\right)$	Method	Boundary Condition	$\Omega_{L}$	$W_{max}^{*}$				
				0.3	0.6	0.9	1.2	1.5
90	Present	CCCC	11.943	1.007	1.026	1.057	1.097	1.154
		SSSS	9.812	1.008	1.032	1.069	1.125	1.198
	DQM [23]	CCCC	11.605	1.005	1.019	1.046	1.087	1.142
		SSSS	9.531	1.003	1.015	1.042	1.089	1.152
180	Present	CCCC	11.189	1.011	1.048	1.110	1.197	1.299
		SSSS	8.764	1.031	1.078	1.147	1.231	1.329
	DQM [23]	CCCC	10.565	1.015	1.058	1.125	1.213	1.319
		SSSS	8.173	1.016	1.061	1.133	1.227	1.341

## Data Availability

Not applicable.

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
