# Peer review of "Large Amplitude Vibration of FG-GPL Reinforced Conical Shell Panels on Elastic Foundation"

_materials, 2023, doi:10.3390/ma16176056_

Round 1
Reviewer 1 Report
Review report
The MS studied the functionally graded (FG) composite structures reinforced by graphene platelets (GPL) have been widely adopted as a state-of-art structural element due to their preeminent properties and functional designability. However, most studies are confined to beams, plates and cylindrical panels relying on the analytical differential quadrature method (DQM) and the finite element numerical method. In this context, current study intends to investigate the nonlinear free vibration of FG-GPL reinforced composite (RC) conical panels resting on an elastic medium by developing a D planar mesh free method-based nonlinear numerical method.
In general, the MS is interested in the elastic foundation topic.
It needs some modifications before considering for publication in Materials Journal as follows:
-Introduction cited Refs. must be put in journal format.
-In Modeling of FG-GPLRC Conical Shell Panel section, the details must considered and finally indicate to Fig. 1 that must put after it directly.
-In page 3, Numbering of Eq. 1 repeated, so the equations numbering must be modified (On other side, author can put one of them 1*)
-Eqs. 1-4 must be cited.
-Vm (z) is not defined.
-Resolution of figure 1 needs clear.
-Eq. 4 needs simplification (i.e., put in one term) if possible.
-Discussion needs some physical meaning of the phenomena in details.
-In conclusion section (A comparison as made in abstract if possible add) to show the new external parameters impact in the current paper.
-All symbols considered put in nomenclature.
Yours Sincerely

English language needs proofreading.
Author Response
Please refer to the Response to reviewers' comments (1) attached.

Reviewer 2 Report
A 2- D planar meshfree method is derived for analysis of the large amplitude vibration of FG-GPL reinforced conical 2 shell panels. Problem considered is complex enough. Paper includes theoretical derivation and numerical analysis. Contribution of the author is obvious.
Remarks
11. „analytical differential quadrature method“. Actually, DQM is commonly known as numerical method, since it is used for numerical solution of differential equations (agree that it include analytical expansions). Thus, it is suggested to replace “analytical method” with “numerical method” or with “DQ discretization method”.
22. In chapter 2 it is correct to add references to formulas taken from literature.
33. „Halphin-Tsai micromechanical model“, short discussion can be added why this model is selected (chapter 3).
44. Proposed approach covers well geometrical non-linearity. Can this approach extended also to non-linear elastic material models? (power law, etc., see doi 10.1007/BF02262803). Are significant changes needed (formulas 15,16). Short discussion can be added regarding possible extension of the proposed approach for nonlinear elasticity. 5. In the case of DQM is important mesh size used, in some table can be shown results obtained with different mesh size (say 8x8 , 16x16).
" And the nonlinear free vibration characteristics of FG-GPLRC conical panels on elastic foundation are profoundly investigated, and it...."
Proofreading suggested.
Author Response
Please refer to the Response to reviewers' comments (2) attached.

Round 2
Reviewer 1 Report
I recommended it for publication in its current form.